# PairwiseNet: Pairwise Collision Distance Learning for High-dof Robot Systems

**Jihwan Kim**
Seoul National University
jihwankim@robotics.snu.ac.kr

**Frank Chongwoo Park**
Seoul National University
fcp@snu.ac.kr

**Abstract:** Motion planning for robot manipulation systems operating in complex environments remains a challenging problem. It requires the evaluation of both the collision distance and its derivative. Owing to its computational complexity, recent studies have attempted to utilize data-driven approaches to learn the collision distance. However, their performance degrades significantly for complicated high-dof systems, such as multi-arm robots. Additionally, the model must be retrained every time the environment undergoes even slight changes. In this paper, we propose PairwiseNet, a model that estimates the minimum distance between two geometric shapes and overcomes many of the limitations of current models. By dividing the problem of global collision distance learning into smaller pairwise sub-problems, PairwiseNet can be used to efficiently calculate the global collision distance. PairwiseNet can be deployed without further modifications or training for any system comprised of the same shape elements (as those in the training dataset). Experiments with multi-arm manipulation systems of various dof indicate that our model achieves significant performance improvements concerning several performance metrics, especially the false positive rate with the collision-free guaranteed threshold. Results further demonstrate that our single trained PairwiseNet model is applicable to all multi-arm systems used in the evaluation. The code is available at https://github.com/kjh6526/PairwiseNet.

**Keywords:** Robot Collision, Collision Distance, Machine Learning

## 1 Introduction

Motion planning algorithms such as RRT [1, 2, 3] and its many variants [4, 5, 6, 7] all require the *collision distance* - the minimum distance between the robot and its nearest obstacle (including other links for self-collision avoidance). Among these, some even require the derivatives of the collision distances. It is well-known that calculating this distance involves finding the minimum distance between each robot link and the obstacles, which can be computationally intensive, especially for high-dof robots with complex geometries.

To alleviate the computational burden, one possible solution is to train a collision distance function using data. By collecting sufficient data consisting of robot configurations and their corresponding collision distances, machine learning models such as kernel perceptron models [8], support vector machines (SVM) [9], and neural networks [10, 11, 12, 13, 14], can be used to learn the collision distance function. This learned function can then be used to quickly determine if a given configuration is collision-free. While these data-driven approaches have demonstrated satisfactory results for low-dof robot systems, often they perform poorly for higher-dof robots. The challenge lies in the fact that the collision distance function for higher-dof robots is complex and highly non-convex.

Another challenge faced by existing data-driven methods is their sensitivity to small environmental changes. For example, the addition of new obstacles or a change of the robot's base position can lead to a completely different collision distance function; for many of these methods, the entire training

7th Conference on Robot Learning (CoRL 2023), Atlanta, USA.

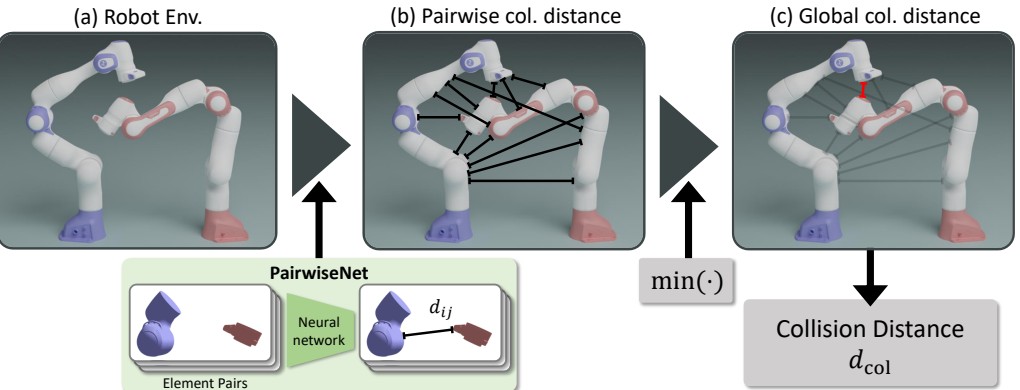

Figure 1: An illustration of the global collision distance estimation through PairwiseNet. (a) Robot environment at a given joint configuration. (b) Pairwise collision distances for all element pairs are determined through PairwiseNet. (c) The smallest of these distances becomes the global collision distance.

procedure must be repeated, from data collection to model training. (One possible exception is [8], which proposes an efficient model update strategy for dynamic environment updates, but their method is limited to low-dof robot systems and still requires an additional training procedure.)

We present PairwiseNet, a collision distance estimation method that provides a promising alternative to existing data-driven approaches used for predicting the *global* collision distance. Instead of directly estimating the global collision distance, PairwiseNet focuses on estimating the *pairwise* collision distance: the minimum distance between two elements in the robot system. The PairwiseNet model takes as input the point cloud data of two geometric shapes and their relative transformation and outputs the minimum distance between these two shapes. To estimate the global collision distance, PairwiseNet first predicts the minimum distances for every possible pair of elements in the system. It then selects the minimum of these pairwise distances as the estimate for the global collision distance (see Figure 1). The efficient parallel batch computation of the neural network enables the rapid prediction of minimum distances between pairs of elements.

Compared to the complex and highly non-convex function of the global collision distance, the minimum distance function between a pair of elements is simpler and easier to train. By breaking down the challenging task of learning the global collision distance into smaller sub-problems of the pairwise collision distance learning, our PairwiseNet achieves significant performance improvements for high-dof robot systems.

Another advantage of PairwiseNet is its applicability to any system composed of known shape elements (shape elements that are sufficiently trained for estimating pairwise collision distance). The trained PairwiseNet model can be used without the need for additional training or modifications in such systems. For example, consider a scenario in which a sufficiently large dataset containing pairwise collision distances between the links of a Panda robot is available. In this case, the PairwiseNet model trained using this dataset can be applied to any system consisting of multiple Panda robots, regardless of the number of robots or their respective positions, as this is possible because the collision distance estimation for such systems can be broken down into pairwise collision distance estimations for each element pair, and these pairwise distances are already known by the trained PairwiseNet model. As long as the system is exclusively comprised of shape elements that have been learned during training, the trained PairwiseNet model is applicable to any such system. Even in cases where the system undergoes changes, such as changing the robot's base position or adding another robot arm, if the geometric shape elements of the system remain unchanged, the trained PairwiseNet model remains applicable to the changed system without any modifications.

Our approach has been evaluated in high-dof multi-arm robot manipulation systems, ranging from the two-arm (14-dof) to four-arm (28-dof) systems, as well as a single-arm robot with obstacles. The results demonstrate that our approach outperforms existing learning-based methods in terms of collision distance regression error, collision checking accuracy, and notably the False Positive Rate with the collision-free guaranteed threshold (Safe-FPR). Moreover, our approach performs better even when using a single trained PairwiseNet model for all multi-arm systems.

## 2 Related Works

Several machine learning-based methods for collision distance estimation have been proposed due to their computationally efficient inference procedures for collision distances and derivatives. [6] used SVM classifiers to identify whether each pair of parts of a humanoid robot was in a safe or dangerous self-collision status given a specific joint configuration. Only the minimum distances of the dangerous pairs of parts were estimated using a capsule-based BV algorithm, simplifying the calculation of collision distances and derivatives. [9] also employed an SVM classifier for a 14-dof dual-arm robot manipulation system. The SVM classifier inputs a vector consisting of the positions of all joints in the system and outputs a collision label of either $<1$ for a collision or $>1$ for a collision-free state. In [13], SVM and neural network models were trained to predict the collision label of a humanoid manipulation system at a given joint configuration. Separate collision classifier models were trained for every sub-part pairs, such as the left arm and right leg, resulting in a total of 10 sub-models used for collision label predictions.

Similar to [9], [10] utilized joint positions as inputs for their multi-layer perceptron neural network model. Meanwhile, [11] employed a positional encoding vector of the joint configuration as input for their neural network model. [12] trained a neural network model to estimate the collision distance using an extended configuration containing both joint and workspace configurations as input, with the model outputting the collision distance of the system. DiffCo [8] is a collision classifier model based on kernel perceptron that generates both the collision score and its derivative. DiffCo also utilizes an efficient active learning strategy that adjusts the trained collision score function for dynamic updates in the environment. Similarly, CollisionGP [15], a Gaussian process-based collision classifier model, has been proposed. CollisionGP determines the collision query for a given joint configuration and also measures the uncertainty of the model in its prediction. Recently, GraphDistNet [14] was proposed as a Graph Neural Network (GNN) model for collision distance estimation. The model inputs the information on geometric shapes, which are represented as graphs for both the manipulator links and obstacles. GraphDistNet then utilizes the geometric relationship between the two graphs to predict the collision distance.

Similar to our method, some works [16, 17] approached the challenge by decomposing a complex problem into several simpler sub-problems. [16] proposed a novel configuration space decomposition method. This method separates the robot into disjoint components and trains a classifier for the collision-free configuration space of each component. Since the components near the base link have a relatively low-dimensional configuration space, training classifiers for these components is easier than training a single classifier for the whole system. [17] trained a collision predictor for generating collision-free human poses. They focused on the fact that collisions only affect local regions of the human body. Therefore, they designed a set of local body parts, and the collision prediction was accordingly decomposed into these local parts.

The effectiveness of these existing methods has been experimentally demonstrated only for low-dof robot systems; their performance degrades substantially for high-dof systems operating in complex environments. In particular, most learning-based collision distance estimation methods establish a collision-free guaranteed threshold to ensure that no collisions occur during actual manipulation. However, existing methods often suffer from a high false positive rate, resulting in overly cautious collision detection when utilizing the collision-free guaranteed threshold. In comparison, our method demonstrates effective collision distance estimation performance even in high-dof robot systems and maintains low false positive rates when using the collision-free guaranteed threshold.

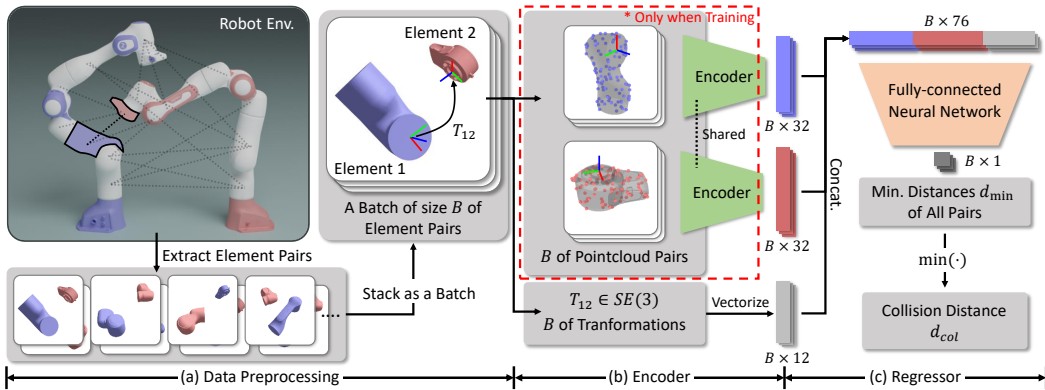

Figure 2: An illustration of estimating the global collision distance via PairwiseNet, our pairwise collision distance learning method.

# 3 Learning Pairwise Collision Distance

## 3.1 Problem Formulation

We assume the availability of a simulator environment of the target system, which includes the robot kinematics and geometric shapes of links and obstacles. We aim to determine the optimal model parameter $\psi$ for the pairwise collision distance estimation model $f_\psi$, which can predict the collision distance between any pair of geometric shapes. The model takes the point cloud data of two geometric shapes $\mathcal{P}_i, \mathcal{P}_j$ (expressed in each corresponding object coordinates) and the relative transformation $T_{ij} \in SE(3)$ as input, and outputs the estimated pairwise collision distance $\hat{d}_{ij}$ between the two shapes.

$$\hat{d}_{ij} = f_\psi(\mathcal{P}_i, \mathcal{P}_j, T_{ij}) \tag{1}$$

After training the model, the global collision distance can be determined by the procedure as shown in Figure 2. First, a set of element pairs and corresponding transformations $\mathcal{S}(q) = \{(\mathcal{P}_i, \mathcal{P}_j, T_{ij}(q))\}_{i,j}$ in the given joint configuration $q$ is extracted from the target robot system. Next, PairwiseNet determines the pairwise collision distance between each element pair in $\mathcal{S}(q)$, and the minimum distance found among these is taken as the global collision distance of the robot system. The global collision distance estimator function $F_\psi$ can be expressed in the form of

$$\hat{d}_{\text{col}}(q) = F_\psi(q; f_\psi, \mathcal{S}) \tag{2}$$
$$= \min_{(\mathcal{P}_i, \mathcal{P}_j, T_{ij}(q)) \in \mathcal{S}(q)} f_\psi(\mathcal{P}_i, \mathcal{P}_j, T_{ij}(q)) \tag{3}$$

where $\hat{d}_{\text{col}}(q)$ is the estimated global collision distance in the joint configuration $q$. Using the batch computation of the neural network model, we can efficiently estimate the minimum distances of element pairs.

## 3.2 Network Architecture

PairwiseNet consists of two main components: an *encoder* that creates a shape feature vector from a point cloud data of a geometric shape (Figure 2b), and a *regressor* that predicts the minimum distance between two shape feature vectors and a transformation (Figure 2c). The encoder employs two EdgeConv layers from Dynamic Graph Convolutional Neural Network [18] to extract 32-dimensional shape feature vectors from the point cloud data. The regressor then combines the two shape feature vectors and the transformation into a single vector and uses four fully connected layers with hidden state dimensions of $(128, 128, 128)$ to output the minimum distance (Figure 2c). The training of PairwiseNet uses the mean-squared error (MSE) between the estimated and actual

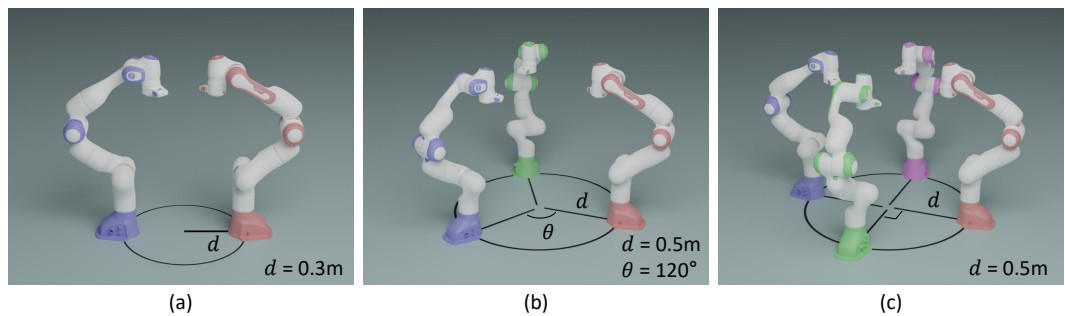

(a)  (b)  (c)

Figure 3: Test environments for the collision distance learning performance evaluation. We selected (a) two arms, (b) three arms, and (c) four arms robot systems.

collision distances as the loss function

$$L = \frac{1}{|\mathcal{D}_{\text{train}}|} \sum_{(\mathcal{P}_i, \mathcal{P}_j, T_{ij}, d_{ij}) \in \mathcal{D}_{\text{train}}} ||f_\psi(\mathcal{P}_i, \mathcal{P}_j, T_{ij}) - d_{ij}||^2 \qquad (4)$$

where $d_{ij} \in \mathbb{R}$ is the ground-truth pairwise collision distance, and $\mathcal{D}_{\text{train}}$ denotes the training dataset.

### 3.3 Efficient Inference Strategy of PairwiseNet

Our approach includes an efficient inference strategy for the global collision distance calculation by eliminating the need to run the encoder, a deep neural network that transforms the point cloud data into feature vectors. Since the point cloud data of element pairs remains unchanged regardless of the joint configuration, shape feature vectors of element pairs can be calculated and saved once for each robot system before calculating the collision distance. Using these pre-calculated shape feature vectors, PairwiseNet is able to estimate the collision distance only using the regressor, a simple neural network composed of fully-connected layers. Implemented in PyTorch [19], PairwiseNet is capable of performing collision distance estimation for the joint configuration in less than 0.5ms. Details on the inference time for PairwiseNet can be found in Appendix B.1.

## 4 Experiments

### 4.1 Collision Distance Learning for Multi-arm Robot Systems

**Target Systems** For the test environments, we selected three high-dof multi-arm robot systems as illustrated in Figure 3. We employed 7-dof Franka Emika Panda robot arms for our test environments. For the test dataset of each target environment, we sampled one million joint configurations from a uniform distribution within the joint limits.

**Training Dataset for PairwiseNet** We collected pairwise collision distance data from dual-arm robot manipulation systems. For the diversity of the dataset, we utilize dual-arm robot systems with various relative positions between the two arms as illustrated in Figure 4. We sample $\theta$ and $\phi$ with eight equally spaced values within the range $[0, 2\pi)$, and sample $R$ with five equally spaced values within the range $[0.1\text{m}, 1.0\text{m}]$. In total, we use 320 different combinations of $(R, \theta, \phi)$, resulting in 320 different dual-arm robot systems.

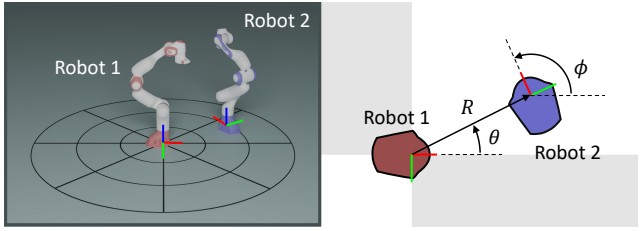

Figure 4: An illustration of multi-arm robot systems for generating the training dataset. Training data points are generated from dual-arm robot environments with various relative positions between two arms.

For each system, we sampled joint configurations uniformly within the joint limits and extracted a set of element pairs $\mathcal{S}(q)$ at each joint configuration $q$ (Figure 2a). To obtain the ground-truth pairwise collision distance $d_{ij} \in \mathbb{R}$ between the element pair, we use the collision distance estimation algorithm implemented in PyBullet [20]. If the two elements collide, the collision distance is the negative of their penetration depth (the distance by which one convex object enters into the interior of another during a collision [21]). The resulting training dataset $\mathcal{D}_{\text{train}}$ contains 3 million data points.

**Baselines**   We trained our method and other existing collision distance estimation methods.

- **Capsule**: A bounding volume method with capsule-shape collision primitives used in [22, 6].
- **JointNN**: A fully-connected neural network model that directly uses joint configurations as inputs (the input representation used in [6, 13]).
- **PosNN**: A fully-connected neural network model that uses joint positions as inputs (the input representation used in [10, 9]).
- **jointNERF** [11]: A fully-connected neural network model that uses positional embedding vectors of joint configurations as inputs.
- **ClearanceNet** [12]: A neural network model that takes joint configurations as inputs and utilizes two fully-connected layers, each followed by a dropout layer.
- **DiffCo** [8]: A kernel perceptron model that takes joint configurations as inputs and outputs the collision score.

Existing collision distance learning methods were trained on one million uniformly sampled data points within the joint limits for each target system. For the DiffCo model, we were limited to a dataset size of 50,000 data points, as this was the maximum feasible size for kernel perceptron training on our hardware (AMD Ryzen Threadripper 3960X, 256GB RAM, and NVIDIA GeForce RTX4090 with 24GB VRAM).

**Performance Evaluation**   We evaluate the performance of collision distance learning using four metrics: MSE, AUROC, Accuracy, and Safe-FPR. These metrics target both the collision distance regression and collision classification, with a robot configuration being classified as a collision if the collision distance is below the threshold $\epsilon$ (for DiffCo, if the collision score is above the threshold). MSE represents the mean squared error between the ground truth and estimated global collision distance. AUROC is the area under the receiver operating characteristic curve for the collision classification. Accuracy is the classification accuracy of collisions with the threshold $\epsilon = 0$. Lastly, in order for the trained collision distance estimation model to be used in actual path planning tasks, a sufficiently conservative threshold must be used to ensure that collisions cannot occur. However, the more conservative the threshold used, the more false alarms will occur where non-collision robot configurations are incorrectly classified as collisions. Safe-FPR is used to evaluate performance in these situations, representing the false alarm rate when using the least yet sufficient conservative threshold that can classify all collision configurations in the test dataset as collisions.

Table 1 presents the evaluation results of PairwiseNet and other existing methods. PairwiseNet outperforms the existing methods in all performance metrics in all three multi-arm robot systems compared to the existing methods, with the top-performing metrics highlighted in bold. Notably, PairwiseNet even shows better performance metrics than Capsule, a computationally expensive BV-based collision distance estimation method.

Figure 5 shows the collision distance estimation results for the four-arm robot system during a collision-free trajectory. The robot configurations at the start and end points of the path are such that they involve all four robot arms intricately intertwined, so the robot arms move through the near-collision region. The ground-truth collision distance and the estimated collision distances by PairwiseNet and other baselines during the trajectory are represented in the bottom-right plot. The performance of PairwiseNet can be confirmed, as it is the only method that accurately estimates the complex ground-truth collision distance of the four-arm robot system.

Table 1: Collision Distance Estimation Performances

| Env. | Methods | MSE | AUROC | Accuracy ($\epsilon = 0$) | safe-FPR |
|------|---------|-----|-------|---------------------------|----------|
| Fig. 3 (a) (Two arms) | Capsule | 5.47e-4 | 0.9995 | 0.9776 | 0.0247 |
| | JointNN | 3.63e-4 | 0.9955 | 0.9794 | 0.3200 |
| | PosNN | 2.71e-4 | 0.9970 | 0.9823 | 0.1476 |
| | jointNERF | 2.98e-4 | 0.9962 | 0.9808 | 0.2371 |
| | ClearanceNet | 1.11e-3 | 0.9853 | 0.9621 | 0.4570 |
| | DiffCo* | - | 0.9824 | 0.9818 | 0.3141 |
| | PairwiseNet (our) | **0.24e-4** | **0.9998** | **0.9941** | **0.0200** |
| Fig. 3 (b) (Three arms) | Capsule | 5.96e-4 | 0.9993 | 0.9775 | 0.0241 |
| | JointNN | 9.69e-4 | 0.9902 | 0.9721 | 0.2679 |
| | PosNN | 5.41e-4 | 0.9951 | 0.9801 | 0.1336 |
| | jointNERF | 8.22e-4 | 0.9920 | 0.9747 | 0.2213 |
| | ClearanceNet | 4.63e-3 | 0.9499 | 0.9395 | 0.6067 |
| | DiffCo* | - | 0.9603 | 0.9453 | 0.5858 |
| | PairwiseNet (our) | **0.24e-4** | **0.9997** | **0.9944** | **0.0189** |
| Fig. 3 (c) (Four arms) | Capsule | 6.66e-4 | 0.9986 | 0.9468 | 0.0694 |
| | JointNN | 1.59e-3 | 0.9718 | 0.9183 | 0.6988 |
| | PosNN | 7.18e-4 | 0.9885 | 0.9478 | 0.5371 |
| | jointNERF | 1.30e-3 | 0.9778 | 0.9280 | 0.6260 |
| | ClearanceNet | 6.67e-3 | 0.8738 | 0.8202 | 0.9965 |
| | DiffCo* | - | 0.8811 | 0.8306 | 0.9874 |
| | PairwiseNet (our) | **0.46e-4** | **0.9994** | **0.9858** | **0.0650** |

*DiffCo outputs the collision score from -1 (collision-free) to 1 (collision).

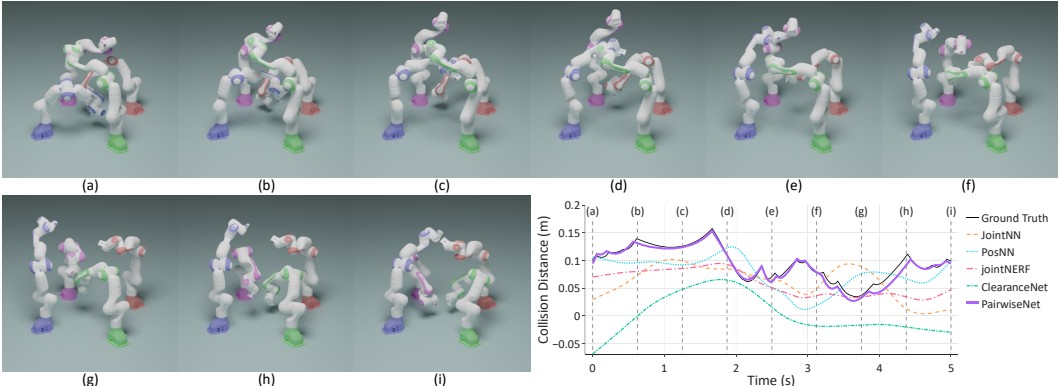

Figure 5: Collision distance estimation for the four-arm robot system: Images depict the four-arm robots following a trajectory, and the plot illustrates ground truth and estimated collision distances of PairwiseNet and other baselines during the trajectory.

**Generalizability** Existing learning methods have used individual models trained from each system's individual dataset to estimate collision distance in the three target robot systems. In contrast, PairwiseNet estimates collision distance with only one model for all three target robot systems, and its performance is even better than that of existing methods that use individual models for each system. Therefore, PairwiseNet, trained with sufficient datasets, can apply the same model without additional training, even if the robot base positions or number of robot arms change.

## 4.2 Collision Detection in a Real-world Environment

We perform experiments in real-world environments with a 7-dof Panda robot arm (Figure 6). This workspace is populated with obstacles such as shelves and tables that add complexity to the robot arm's operational landscape. To validate the performance, we provide a human-guided demonstration with the robot arm occasionally sweeping close to, and sometimes colliding with, tables and shelves.

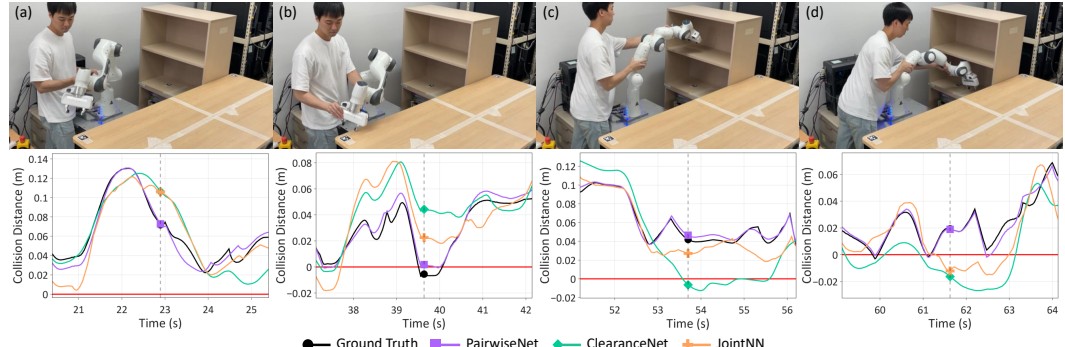

Figure 6: Collision distance estimation for the 7-dof Panda robot arm environment with obstacles (tables and shelves). The top images display a human-guided robot arm, while the corresponding plots at the bottom illustrate the ground truth and estimated collision distances from PairwiseNet and other baselines at time $t =$ (a) 22.9s, (b) 39.6s, (c) 53.7s, and (d) 61.6s, respectively.

The training procedure of PairwiseNet and other baselines, as well as the dataset and model architectures used, are identical to those employed in the experiments for multi-arm robot systems. In the case of PairwiseNet, the dataset consists of element pairs representing tables and shelves, which are further divided into sub-parts. Specifically, a table is divided into a tabletop and four legs, while a shelf is divided into six plates. The pairwise collision distances between these sub-parts and the robot links are calculated by PyBullet for the training dataset.

The collision distance estimation results are presented in Figure 6. Note that Figure 6 displays only four snapshots from the demonstration; the complete video can be accessed in the supplementary [1]video. Compared to the evaluated baselines, PairwiseNet consistently exhibits the highest accuracy in estimating the actual collision distance. It reliably detects collisions between the robot arm and obstacles in the majority of cases. In contrast, the other baselines either fail to trigger collision alarms when a collision occurs (Figure 6 (b)) or produce false collision alarms when no collision is present (Figure 6 (c), (d)). PairwiseNet stands out by consistently and accurately identifying collisions between the robot and obstacles.

## 5 Conclusions

In this paper, we present PairwiseNet, a novel collision distance estimation method that estimates the minimum distance between a pair of elements instead of directly predicting the global collision distance of the robot system. By simplifying the problem into smaller sub-problems, our approach achieves significant performance improvements for high-dof robot systems compared to methods that directly predict the global collision distance. Additionally, PairwiseNet is capable of handling environmental changes such as robot base repositioning without requiring additional training or fine-tuning. We evaluate and compare the collision distance estimation performance of PairwiseNet for both high-dof multi-arm robot systems and single-arm systems in the presence of obstacles, and validate its accurate collision distance estimation and generalization to environmental changes.

**Limitations and Future works**   The generalizability of PairwiseNet is currently limited to systems that exclusively consist of known shape elements. While PairwiseNet demonstrates robust performance in estimating collision distances within this scope, its applicability to environments with unknown (untrained) shape elements has not been fully investigated. Future work is aimed at enhancing the generalizability of PairwiseNet to accommodate systems that include previously unseen shape elements. This could involve, e.g., expanding the training dataset to incorporate a wider range of environmental variations, and incorporating techniques to handle unknown or novel shape elements [23].

---

[1]https://youtu.be/N5Q8ZXbB6Uc

**Acknowledgments**

This work was supported in part by IITP-MSIT grant 2021-0-02068 (SNU AI Innovation Hub), IITP-MSIT grant 2022-0-00480 (Training and Inference Methods for Goal-Oriented AI Agents), KIAT grant P0020536 (HRD Program for Industrial Innovation), ATC+ MOTIE Technology Innovation Program grant 20008547, SRRC NRF grant RS-2023-00208052, SNU-AIIS, SNU-IAMD, SNU BK21+ Program in Mechanical Engineering, and SNU Institute for Engineering Research.

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

# Appendix

## A    Further Experimental Details

### A.1    Strategy for Decomposing System Elements

PairwiseNet achieves robust collision distance estimation performance by dividing the robot system into multiple elements and calculating the pairwise collision distance between these elements. Therefore, to apply PairwiseNet, the robot system must be divided into these elements as a preliminary step. Each element must be a rigid body with an unchanging shape, and in the case of the Panda robot arm used in our experiments, each link was treated as a separate element. Since PairwiseNet does not require each element to be convex, non-convex links of the Panda robot arm can be used without additional decomposition.

However, PairwiseNet's superior performance is attributed to the fact that the pairwise collision distance functions between elements are much easier to learn than the global collision distance function. Therefore, if an individual element's shape is complex and highly non-convex, learning the corresponding pairwise collision distance may become difficult, potentially diminishing PairwiseNet's effectiveness. Hence, a well-considered balance must be found between the complexity of decomposing the system and the performance of PairwiseNet.

In our real robot experiments, we decomposed tables into a tabletop and four legs, and shelves into six plates. This strategy was employed not only to simplify complex obstacles and make the pairwise collision distance more tractable but also to leverage the symmetry of the obstacle structure. For example, the bottom, middle, and top plates of the shelf have the same shape, so dividing them into separate elements allows for more efficient use of training data. Furthermore, the fact that the decomposed elements of the tables and shelves take the form of flat rectangular shapes can be beneficial to the learning process.

### A.2    Capsule-based Bounding Volume Method for a Panda Robot Arm

We construct capsule-shaped collision primitives for a Panda robot arm as a baseline collision distance estimation method (see Figure 7). Our approach follows a similar methodology of [22], which formulates an optimization problem as follows:

$$\min_{a_i,b_i,r_i} \quad ||a_i - b_i||\pi r_i^2 + \frac{4}{3}\pi r_i^3 \tag{5}$$

$$\text{s.t.} \quad \text{dist}(p, \overline{a_ib_i}) \leq r_i, \quad \text{for all } p \in \mathcal{M}_i \tag{6}$$

Here, $i$ denotes the link index of the Panda robot arm, $\mathcal{M}_i$ represents the vertices of the $i^{\text{th}}$ link mesh, and $a_i$, $b_i$, and $r_i$ refer to the two endpoints and the radius of a capsule, respectively, and $\overline{a_ib_i}$ represents the line segment connecting the two endpoints. This formulation results in the creation of minimal volume capsules that encapsulate all vertices of the link meshes. The collision distance of the multi-arm robot systems can be estimated through the minimum distance calculation between capsules.

### A.3    The Collision-free Guaranteed Threshold

The collision-free guaranteed threshold $\epsilon_{\text{safe}}$ refers to a predefined distance value that is established in collision distance estimation methods. This threshold is set to ensure that during testing or actual operation, the estimated collision distance remains above this threshold for all valid configurations or movements of the robot system. In other words, if the estimated collision distance between the robot and any obstacles remains above the collision-free guaranteed threshold ($\hat{d}_{\text{col}}(q) > \epsilon_{\text{safe}}$), it is considered safe and collision-free. In our experiments, we set the collision-free guaranteed threshold to the least conservative value that allows us to classify all the collision configurations in the test dataset as collisions. These thresholds are then utilized for measuring the Safe-FPR.

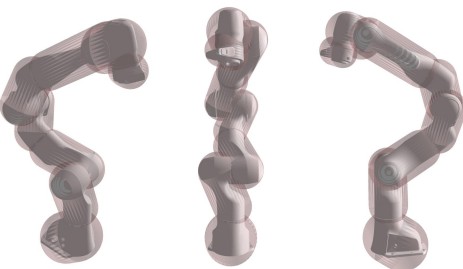

Figure 7: Illustrations of the Panda robot arm with capsule-shape collision primitives.

Table 2: The Collision-free Guaranteed Thresholds

| Methods | Two arms | Three arms | Four arms |
|---|---|---|---|
| Capsule | 0.0 | 0.0 | 0.0 |
| JointNN | 0.2111 | 0.2015 | 0.2231 |
| PosNN | 0.1141 | 0.1189 | 0.1756 |
| jointNERF | 0.1661 | 0.1734 | 0.2001 |
| ClearanceNet | 0.2840 | 0.3713 | 0.4944 |
| DiffCo | -1.2789 | -1.4672 | -0.9535 |
| PairwiseNet | 0.0150 | 0.0152 | 0.0184 |

## A.4 Hyperparameters

Table 3 shows hyperparameters employed in our experiments.

Table 3: Hyperparameters

| hyperparameter | value |
|---|---|
| batch size, learning rate, epoch for PairwiseNet | 1000, 1e-3, 2000 |
| batch size, learning rate, epoch for ClearanceNet | 191, 1.75e-4, 400 |
| batch size, learning rate, epoch for other NN baselines | 10000, 1e-3, 10000 |
| $k$ for $k$-nearest neighbor of EdgeConv layers | 5 |
| # of points in the point cloud data of an shape element | 100 |
| hidden nodes of EdgeConv layers | 64 |

# B    Additional Experimental Results

## B.1    Inference Time Comparison between PairwiseNet and a Standard Collision Distance Estimation Method

PairwiseNet incorporates an efficient inference strategy – using only the regressor network during the inference process, and computing multiple element pairs as a single batch. Thus, despite calculating collision distances pairwise like traditional non-data-driven methods, it demonstrates an inference speed as fast as other existing data-driven approaches. We have compared the inference speed of PairwiseNet with that of standard non-data-driven methods and displayed the results in Table 4. The standard collision distance calculation algorithms used for comparison are from the Flexible Collision Library (FCL) and its extended implementation (HPP-FCL).

The test environments were the same three multi-arm robot systems used to evaluate the collision distance estimation performance. The Panda robot arm provides a simplified convex mesh, representing the original link mesh with only 1/60 of the number of vertices, allowing for efficient collision distance estimation. In our experiments, we measured the inference time using both the original complex mesh and the simplified mesh. The inference time was measured as the time taken to estimate the collision distance for a total of 1,000 joint configurations.

Table 4: Inference Time Comparison

| Methods | Inference time for 1000 joint poses (s) | | | | | |
| --- | --- | --- | --- | --- | --- | --- |
| | Two arms (64 pairs) | | Three arms (192 pairs) | | Four arms (384 pairs) | |
| FCL w/ original mesh | 13.87 | | 43.97 | | 92.63 | |
| FCL w/ simplified mesh | 2.479 | | 9.079 | | 17.25 | |
| HPP-FCL w/ simplified mesh | 0.3920 | | 0.8560 | | 1.9490 | |
| | CPU | GPU | CPU | GPU | CPU | GPU |
| ClearanceNet | 0.1450 | 0.0919 | 0.1484 | 0.0818 | 0.1493 | 0.0825 |
| ClearanceNet (Batch) | 0.0299 | 0.0001 | 0.0250 | 0.0001 | 0.0222 | 0.0001 |
| PairwiseNet | 0.1054 | 0.0988 | 0.1590 | 0.0998 | 0.2025 | 0.1045 |
| PairwiseNet (Batch) | 0.0362 | 0.0207 | 0.1070 | 0.0224 | 0.2121 | 0.0253 |

Additionally, PairwiseNet is capable of receiving multiple joint configurations as a single batch input; it can simultaneously compute the collision distances for all the joint configurations. This is an efficient feature for tasks that require simultaneous collision checks for multiple joint configurations, such as sampling-based path planning methods like RRT or control techniques such as Model Predictive Control (MPC). In our experiments, we also measured the time taken to calculate the collision distances for 1,000 joint configurations at once through PairwiseNet, and this is represented under the PairwiseNet (Batch) entry in Table 4. PairwiseNet was implemented using PyTorch [19]. FCL and HPP-FCL were implemented using the `python-fcl` and `hpp-fcl` libraries, respectively. Thus, all the time measurements are conducted within Python code. The experiments were carried out in an environment equipped with an AMD Ryzen 9 7950X (16 cores, 32 threads), NVIDIA RTX 4090, and 125GB of RAM.

PairwiseNet demonstrates an inference speed that is at least 20 times and up to more than 150 times faster than FCL. As the number of element pairs requiring collision distance calculation increases, the gap between the two widens. This is because in the case of FCL, the time taken for inference increases as the number of element pairs grows, whereas, with PairwiseNet, there is little change in inference time even as the number of pairs increases. HPP-FCL is shown to be 6~9 times faster than the original FCL. However, PairwiseNet still demonstrates an inference speed that is 4~18 times faster than HPP-FCL. Moreover, the inference time of HPP-FCL increases as the number of pairs increases, which is not the case with PairwiseNet. Even when performing inference through a CPU, the inference time does not increase significantly. For batch calculation, where 1,000 joint configurations are computed at once, PairwiseNet can estimate 1,000 collision distances in as little as 25 milliseconds.

## B.2 Training Complexity of PairwiseNet

We utilized 3 million data points (3 million shape element pairs with their corresponding distances) to train PairwiseNet for collision distance estimation in multi-arm robot systems and for single-arm systems with obstacles. For a more detailed description of the complexity of PairwiseNet's training process, additional information is presented in Table 5.

- *Unique element pairs* refers to the count of element pairs in the system, excluding those with duplicate shapes (for example, the pair of the robot arm's seventh link with the top shelf plate and the pair with the middle shelf plate are considered the same since the shapes of the top and middle plates are identical). The more unique element pairs, the greater the number of pairwise collision distances PairwiseNet must learn, resulting in higher training complexity.

- *Gradient step* refers to the number of times the learnable parameters were updated to minimize the loss during the training process.

- *Training time elapsed* refers to the total time taken to complete the training.

- The table also includes *Validation loss* to represent each training result.

The training time was measured on an environment with AMD Ryzen 9 7950X (16 cores, 32 threads), NVIDIA RTX 4090, and 125GB of RAM environment.

Table 5: Training Complexity of PairwiseNet

| Training Env. | Unique element pairs | Training data points | Gradient steps | Training time elapsed (h) | Validation loss (MSE) |
|---|---|---|---|---|---|
| Multi-arm | 36 | 1,000,000 | 2,860,000 | 24.8 | 1.46e-5 |
| Multi-arm | 36 | 3,000,000 | 4,286,000 | 35.6 | 1.43e-5 |
| Single arm w/ obstacles | 70 | 3,000,000 | 4,286,000 | 36.1 | 8.03e-6 |
| Two arms w/ obstacles | 64 | 1,000,000 | 2,860,000 | 23.3 | 9.82e-6 |

We conducted additional experiments to analyze the training complexity of PairwiseNet. Initially, for the existing multi-arm robot systems, while we began with 3 million data points for training PairwiseNet, we also conducted experiments using fewer data points (1 million) and fewer gradient steps. While the training time decreased, the learning results were comparable to those with the original 3 million data points.

Next, we examined the previously mentioned single-arm system with obstacles. Although there were many unique element pairs (70), since all obstacle elements were rectangular, they were relatively easier to learn. Additionally, we trained PairwiseNet with a two-arm robot system, adding non-rectangular household objects as obstacles (as shown in Figure 8). Despite using merely a total of 1 million data points and fewer gradient steps in this scenario compared to the original PairwiseNet, the validation loss was successfully reduced to 9.82e-6, confirming successful learning.

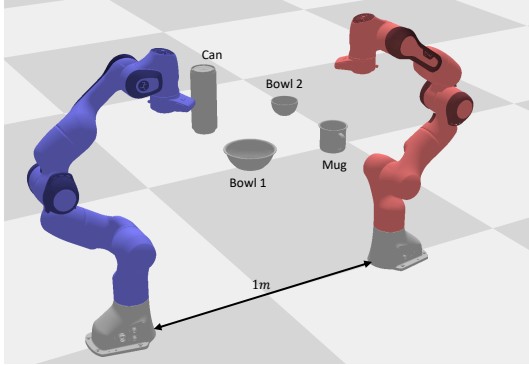

Figure 8: A two-arm robot system with four household objects

## B.3 Evaluating PairwiseNet in Multi-arm Robot Systems with Various Base Positions

We extended our validation of PairwiseNet to include robot arm systems arranged asymmetrically and in an irregular manner, in addition to the three multi-arm robot systems used in our experiments. These additional systems are illustrated in Figure 9. In each systems, the robot arms are arranged in a more irregular and complex manner than in the previously used systems, resulting in a greater variety of relative positional relationships between the arms.

We used the trained PairwiseNet model that was originally used for our multi-arm robot systems. Although these new robot systems have complex arrangements of robot arms, they still consist of Panda robot arms, which have been sufficiently trained. Therefore, PairwiseNet can be applied directly to these systems without the need for additional training.

We have presented the results of PairwiseNet's collision distance estimation performance in Table 6. Upon examining these results, we observed that PairwiseNet performed well across all robot systems and metrics, regardless of the complexity of the arrangement of robot arms. The consistency in performance across these varying configurations demonstrates the robustness of PairwiseNet, af-

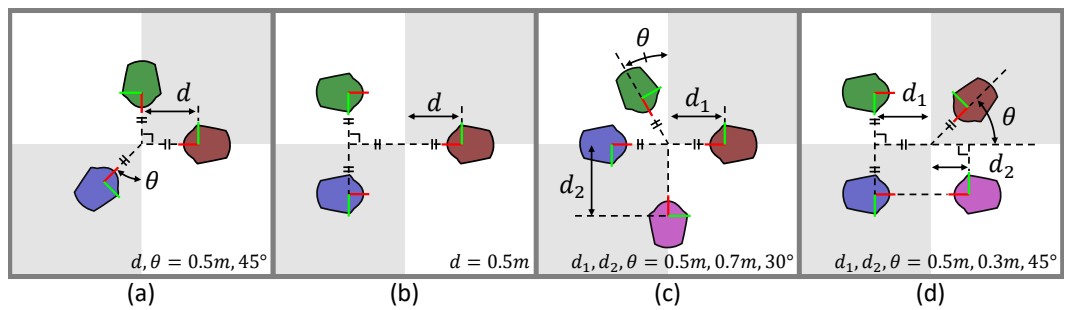

Figure 9: Illustrations of the top views of various base positions within multi-arm robot systems.

firming that there is no significant difference in the quality of collision distance estimation between these different scenarios.

Table 6: Collision distance estimation performances of PairwiseNet for various multi-arm systems

| Env. | MSE | AUROC | Accuracy ($\epsilon = 0$) | safe-FPR |
|---|---|---|---|---|
| Fig. 9 (a) (Three arms) | 0.24e-4 | 0.9997 | 0.9943 | 0.0341 |
| Fig. 9 (b) (Three arms) | 0.17e-4 | 0.9999 | 0.9987 | 0.0048 |
| Fig. 9 (c) (Four arms) | 0.43e-4 | 0.9991 | 0.9859 | 0.0611 |
| Fig. 9 (d) (Four arms) | 0.27e-4 | 0.9995 | 0.9931 | 0.0292 |

## B.4 Comparing the Scalability to High-dof Systems of PairwiseNet with GraphDistNet

Without directly comparing the collision distance estimation performance, we can highlight the novelty of PairwiseNet relative to GraphDistNet [14]. A key difference lies in scalability to high-dof systems. GraphDistNet's performance was only verified in systems with a maximum of 7-dof, and that was within the synthetic planar robot systems where simple graph representations were possible. Additionally, experiments with real robots were only conducted with 3-dof. Since GraphDistNet employs a complex GNN structure that takes the graph itself as the input, the model's inference speed is inevitably sensitive to the complexity of the graph. In examining the experimental results, we observed that the inference speed slows down as the graph becomes more complex, being up to 120 times slower than ClearanceNet [12]. In contrast, PairwiseNet maintains inference speeds comparable to ClearanceNet, even in complex systems with 28 DOF across 4 robot arms.

