# OpenReview forum: "PairwiseNet: Pairwise Collision Distance Learning for High-dof Robot Systems"
_robot-learning.org/CoRL/2023/Conference — CoRL 2023 Poster_

### Official Review · Reviewer_vQwN · 2023-07-17

**Confidence:** 4
**Originality:** Good
**Technical Quality:** Good
**Clarity Of Presentation:** Very Good
**Impact:** 3

**Recommendation:**

Weak Accept: I recommend accepting the paper, but will not argue for my recommendation if the majority of other reviewers have a different opinion.

**Review:**

Strengths:

- Using a neural network to encode two known shapes with any transformation for distance is interesting. It is efficient and can be extended to high DOF robots.
- The experiments show the efficiency and effectiveness of the method.

Weaknesses:

- The training complexity of this method may increase quadratically as DOF increases, e.g. every link pair of a humanoid robot will be considered
- Each robot link has relatively simple geometry and may be represented by several convex shapes. Real-world experiments also decomposite obstacles to several convex shapes.

I think this paper's idea is interesting, decompose collision checking into subproblems and consider using a neural network to encode the relative transformation of shapes and get their distance. However, by this kind of decomposition, the shapes are nearly convex. I am curious what is the performance of directly computing the cage (maybe a combination of several convex shapes) of the shape and using GJK to compute the shapes' distance after transformation.

**Quality Of The Limitations Section:**

Additional details required

**Questions For Rebuttal:**

I am interested in one additional experiment. What is the performance of directly computing the cage (maybe a combination of several convex shapes) of the shape and using GJK to compute the shapes' distance after transformation?

**Robotics Focus:**

Sufficient demonstration on hardware

**Summary Of Paper:**

The paper proposes a method that views collision detection of articulated bodies as the computation of the distance between the robot's links pairs with a transformation. Once trained, this method is able to extend to high DOF and multi-robot scenes. Experiments also show the efficiency and effectiveness of the method.

**Summary Of Recommendation:**

I think this paper's idea is interesting. However, after the problem decomposition, I think the subproblem may be easily solved by traditional methods.

---

> ### Author Response · Authors · 2023-08-12
> **Response to Reviewer vQwN (2/2)**
>
> **Q3.** I am interested in one additional experiment. What is the performance of directly computing the cage (maybe a combination of several convex shapes) of the shape and using GJK to compute the shapes' distance after transformation?
>
> **A3.** Thank you for your insightful comment and interest in additional experiments. In our research, we used Franka Emika Panda robot arms that provide simplified convex meshes to cover the original complex meshes of the link. This design is for efficient collision checking, as the simplified meshes were constructed with only 1/60 of the number of vertices found in the original meshes. We utilized FCL (Flexible Collision Library) and calculated the MSE between the collision distances of 1000 joint configurations from the original and simplified meshes, which amounted to $1.2e^{-4}$. This error can be considered allowable for high-level tasks such as path planning; however, the primary concern is the computation speed of the standard collision distance estimation method. `We have added an inference time comparison between PairwiseNet and FCL in Appendix B.1.` PairwiseNet demonstrates a minimum of 20 to a maximum of 150 times faster inference speed than FCL. Moreover, FCL becomes progressively slower as the number of pairs increases, a drawback not encountered with PairwiseNet. Even in the use of a CPU, PairwiseNet outperforms FCL in terms of inference speed. PairwiseNet also holds additional advantages such as (1) the ability to infer the collision distances of multiple joint configurations at once, and (2) the efficient calculation of the derivative of the collision distance, which proves to be useful for optimization-based path planning methods. In summary, while using simplified meshes as cages for distance calculation can be efficient in certain contexts, our results emphasize the robustness and efficiency of PairwiseNet as a superior approach. Thank you once again for your comment, as it has allowed us to further clarify the novelty of our work and emphasize the unique advantages of PairwiseNet.

---

> > ### Comment · Reviewer_vQwN · 2023-08-13
> > **Thanks for your response**
> >
> > Thanks for your response and experiment on FCL. However, as far as I know, HPP-FCL provides a more efficient GJK algorithm (https://github.com/humanoid-path-planner/hpp-fcl), where one comparison uses around 1 microsecond and two arm pair case may use around 0.064 seconds. I think the learning-based method is good for a big batch comparison. I am not sure if there exists GPU-based GJK for batch collision detection, if not, this method shows some advantages and I am willing to accept it.

---

> > > ### Author Response · Authors · 2023-08-14
> > > **Response to Reviewer vQwN**
> > >
> > > Thank you for your valuable suggestion. In response, we implemented collision distance estimation based on HPP-FCL and compared the inference time with that of the original FCL and PairwiseNet. The comparison results are presented in the table below, and we planned to add this result in Appendix B.1 for the camera-ready version:
> > >
> > > |           Methods          | Two arms | Three arms | Four arms |
> > > |:--------------------------|:--------:|:----------:|:---------:|
> > > | FCL w/ simplified mesh     |  2.479s  |   9.079s   |   17.25s  |
> > > | HPP-FCL w/ simplified mesh |  0.3920s  |   0.8560s   |   1.9490s  |
> > > | PairwiseNet (GPU)          |  0.0988s |   0.0998s  |  0.1045s  |
> > > | PairwiseNet (GPU, Batch)   |  0.0207s |   0.0224s  |  0.0253s  |
> > >
> > > As can be seen from the table, HPP-FCL shows that it is 6\~9 times faster than the original FCL. However, PairwiseNet still demonstrates an inference speed that is 4\~18 times faster than HPP-FCL. Moreover, the inference time of HPP-FCL increases as the number of pairs increases, which is not the case with PairwiseNet.
> > >
> > > As you mentioned, one of the novelties of PairwiseNet is its ability to perform batch collision distance estimation. The inference time for 1000 joint configurations via batch calculation with PairwiseNet takes as little as 0.0253s, which is only 1/77 of HPP-FCL's inference time. This performance gap increases as the number of joint configurations increases. As far as we can ascertain, no prior research has been conducted on performing batch collision checking with GJK. ([12] also highlights the ability of their neural network-based collision distance estimation model to perform batch collision checking, in comparison to traditional methods.)
> > >
> > > In summary, PairwiseNet exhibits remarkable collision distance estimation performance while having the ability to efficiently perform batch collision distance estimation. Moreover, PairwiseNet can provide the derivative of the collision distance with respect to the joint configuration, even for a batch of joint configurations. Once again, thank you for your consideration.

---

> ### Author Response · Authors · 2023-08-12
> **Response to Reviewer vQwN (1/2)**
>
> **Q1.** The training complexity of this method may increase quadratically as DOF increases, e.g. every link pair of a humanoid robot will be considered.
>
> **A1.** Thank you for bringing up the important concern regarding the training complexity of our method, especially as the degrees of freedom (DOF) increases. You are correct in pointing out that the required data points for the training dataset may increase with an increase in DOF, as every link pair and its pairwise distance should be sufficiently included in the dataset. This indeed poses a challenge that we are aware of.  `We have added Appendix B.3 for addressing this issue.` The scalability of the methodology is a critical issue that must be considered for real-world applications, and it is indeed a necessary aspect for PairwiseNet as well. One possible approach is the use of transfer learning, employing a pre-trained network as the encoder to extract information about the element shape. Since the encoder network is not used during the inference process, its structure and size can be flexible. Thus, a trained network that extracts features from point cloud data can be initialized as the encoder to explore collision distance learning. As this approach uses meaningful shape features from the very beginning of the learning process, we can anticipate a faster learning speed compared to the conventional training procedure. Thanks to your comment pointing out this area, we have gained insight into the direction we need to explore for the further development of PairwiseNet. Once again, we thank you for your insightful comment.
>
> **Q2.** Each robot link has relatively simple geometry and may be represented by several convex shapes. Real-world experiments also decomposite obstacles to several convex shapes.
>
> **A2.** Thank you for the insightful comment. You are correct in stating that each robot link can often be represented by several convex shapes, and that real-world experiments also tend to decompose obstacles into several convex shapes. It's important to note that PairwiseNet does not constrain the element's convexity, meaning that the representation of shapes is not limited to convex forms. However, PairwiseNet's novel performance mainly arises from the simplicity of the pairwise collision distance function as compared to the global collision function. Should a single element be complex and exhibit some non-convexity, then the pairwise collision distance function might become more intricate, potentially degrading the performance of PairwiseNet. Recall that PairwiseNet does not constrain the element's convexity, so it is worth exploring in future works to determine the maximum complexity of a single element for which PairwiseNet can successfully learn the pairwise collision distance function. `We have added these discussions about decomposing the shape elements to Appendix A.1.` Again, I appreciate your comment, which gives us valuable insight and directions for future research. Your observation could lead to further refinements and enhancements of PairwiseNet, and we are thankful for your thoughtful contribution.

---

### Official Review · Reviewer_fxzY · 2023-07-19

**Confidence:** 5
**Originality:** Good
**Technical Quality:** Fair
**Clarity Of Presentation:** Very Good
**Impact:** 3

**Recommendation:**

Strong Reject: I recommend rejecting the paper and will argue for my recommendation even if other reviewers hold a different opinion.

**Review:**

Strengths – The main strength of this paper is the decomposition of the distance estimation with complex geometry into parallel sub-problems.

Weaknesses – The method requires known geometries that cannot be scalable and generalizable to new setups. Further, there is no comparison with conventional/representative distance estimation methods such as the GJK method.

This paper needs to clarify and improve a few things:
- A major concern is the scope of this work and its limitation. This work assumes the known geometry of robots. The pairwise distance estimation seems to be a good idea, but current distance computation already performs in the same manner. Further, the existing algorithms such as GJK algorithm, which has been used in Flexible Collision Library, are quite efficient if simple geometries are given. Unlike the proposed method, the other methods are scalable/generalizable to variants of geometries that can be given by sensors. Thus, although this proposed method shows efficiency in the defined setup, it is hard to know if we really need this without further evaluation with other distance computation methods.


**Quality Of The Limitations Section:**

Additional details required

**Questions For Rebuttal:**

-	This paper needs to show the proposed method is faster and/or more accurate than the non-data-driven algorithm such as the GJK method.
-	 L182, the penetration depth is hard to define depending on the shape of the geometries. What is the definition of penetration depth?
-	Most methods resulted in good performance. How significant result does PairwiseNet make?


**Robotics Focus:**

Sufficient demonstration on hardware

**Summary Of Paper:**

This paper aims to develop a data-driven collision-distance estimator that finds the distance between multiple manipulators. However, this is challenging due to the complexity of high-degree-of-freedom robots and the environmental variabilities. To figure it out, this paper proposes PairwiseNet, a collision distance estimation method that finds the pair-wise distance between two sub and known geometries of robots. The main contributions of this paper is as follow:
-	Efficient distance estimation formulating easily solvable smaller sub-problems
-	No need for additional training or modification


**Summary Of Recommendation:**

The proposed methodology shows good accuracy in the setup. However, the method requires strong assumptions and missing comparisons with major methods in the field. I would like to recommend improving this paper.

---

> ### Author Response · Authors · 2023-08-12
> **Response to Reviewer fxzY (2/2)**
>
> **Q2.** L182, the penetration depth is hard to define depending on the shape of the geometries. What is the definition of penetration depth?
>
> **A2.** We apologize for not providing the definition of penetration depth in the original manuscript. The penetration depth, in the context of our work, is determined using the Expanding Polytope Algorithm (EPA). The Expanding Polytope Algorithm (EPA) is often used in conjunction with the GJK algorithm to find the penetration depth when two convex shapes intersect. Penetration depth is the distance by which one object enters into the interior of another object during a collision. The EPA calculates this by expanding a polytope within the Minkowski difference of the two intersecting shapes, focusing on the face closest to the origin. When the polytope can no longer be expanded, the distance from the closest face to the origin represents the penetration depth. This definition holds for various geometrical shapes, as long as they are convex. For a more detailed explanation, please refer to Van Den Bergen, Gino. "Proximity queries and penetration depth computation on 3D game objects." Game developers conference. Vol. 170. 2001. `We have clarified this definition in the revised manuscript.`
>
> **Q3.** Most methods resulted in good performance. How significant result does PairwiseNet make?
>
> **A3.** Thank you for your comments, and we apologize for not sufficiently emphasizing the novelty of the methodology presented in our manuscript. As you pointed out, looking at the collision distance estimation performance in Table 1, most of the methods show over 90% accuracy in collision checking performance and we agree that the improvement in PairwiseNet's performance might not seem evident.
>
> We considered the collision distance estimation performance as two distinct problems: collision classification and regression of collision distances, and we selected performance metrics accordingly. First, the collision classification problem measures the accuracy of distinguishing between collision and non-collision states. This is done by setting a threshold for the estimated collision distance and classifying it as a collision if the distance is lower than that threshold. For this classification problem, we used the widely-accepted performance metrics AUROC and Accuracy. PairwiseNet shows the best performance in both of these metrics; however, as mentioned, most methods demonstrate over 90% accuracy in collision classification making PairwiseNet's improvement appear less significant.
>
> For this reason, we additionally used a performance metric called safe-FPR. When using a collision distance estimation algorithm for real-world path planning or robot control, safety considerations require limiting False Negatives, i.e., classifying actual collisions as non-collisions. To achieve this, a conservative threshold for collision distance must be set. However, a more conservative threshold increases the risk of False Positives or false alarms (classifying non-collisions as collisions), which can significantly reduce robot productivity. Safe-FPR measures the frequency of these false alarms when a sufficiently conservative threshold is applied. Notably, existing learning-based methods exhibit a high safe-FPR, classifying between 20% and 90% of non-collisions as collisions, thereby generating many false alarms when a conservative threshold is used. In contrast, PairwiseNet maintains a low safe-FPR (2% to 6.5%), comparable even to the non-data-driven capsule bounding volume method for estimating collision distance.
>
> Next, we evaluate the regression performance through the Mean Square Error (MSE), which measures the discrepancy between the actual collision distance and the estimated collision distance. PairwiseNet's MSE demonstrates outstanding performance, being about 1/20 of the MSE of other methods. Notably, it even surpasses the capsule bounding volume method, showing a significantly lower MSE. PairwiseNet's impressive ability in tracking collision distance can be further verified through the graph in Figure 5 and the video included in the Supplementary materials. It is unique in successfully tracking the complex and highly non-convex collision distance within a multi-robot arm system.
>
> In summary, PairwiseNet excels in performance compared to existing learning-based methods, particularly in systems with high degrees of freedom. This advantage can translate into significant benefits for high-level tasks that require collision distance calculations, such as path planning. Once again, we thank you for highlighting this vital aspect of performance metric comparison. Your insight has enabled us to further clarify our performance metrics and the significance of the differences in performance.

---

> ### Author Response · Authors · 2023-08-12
> **Response to Reviewer fxzY (1/2)**
>
> **Q1.** A major concern is the scope of this work and its limitation. This work assumes the known geometry of robots. The pairwise distance estimation seems to be a good idea, but current distance computation already performs in the same manner. Further, the existing algorithms such as GJK algorithm, which has been used in Flexible Collision Library, are quite efficient if simple geometries are given. Unlike the proposed method, the other methods are scalable/generalizable to variants of geometries that can be given by sensors. Thus, although this proposed method shows efficiency in the defined setup, it is hard to know if we really need this without further evaluation with other distance computation methods. ... This paper needs to show the proposed method is faster and/or more accurate than the non-data-driven algorithm such as the GJK method.
>
> **A1.** We are thankful for your thoughtful comments and concerns regarding our work. As you have pointed out, our methodology does assume the known geometric shape of robots and obstacles, calculating the collision distance in a pairwise manner. Therefore, we understand your observation that PairwiseNet may not appear novel when compared to existing standard collision distance calculation methods, such as GJK.
>
> However, to the best of our knowledge, calculating collision distance using existing non-data-driven methods can still be computationally expensive. This challenge is particularly pronounced in systems with high degrees of freedom, such as humanoids or multi-arm systems. Many recent studies have already presented data-driven-based collision distance calculation methods to address these problems. (Most of these studies are cited in our manuscript for your reference.) These proposed data-driven methods demonstrate remarkably faster collision checking speeds compared to traditional non-learning methodologies. Additionally, they can efficiently provide the derivative of the collision distance with respect to joint configurations, making them highly valuable in various path planning tasks.
>
> The issue is that these existing data-driven methods often fall short in systems with high degrees of freedom. PairwiseNet distinguishes itself from these methods by inferring collision distances in a pairwise manner, thus demonstrating excellent performance even in systems with high degrees of freedom. At the same time, PairwiseNet inherits the advantages of data-driven approaches (such as rapid collision distance and derivative inference) through its efficient inference strategy, which involves distance inference using only a regressor, and through the efficient batch computation capability of the neural network.
>
> `We have added results in Appendix B.1 that compare the distance inference speed of PairwiseNet with a standard non-data-driven algorithm (FCL).` In our measurements, which involved estimating the collision distance for 1000 joint configurations, PairwiseNet showed an inference speed that is 20 to over 100 times faster than FCL. Furthermore, PairwiseNet has the ability to process numerous joint postures simultaneously in a batch, allowing for the rapid calculation of collision distances for as many as 1000 joint postures in a brief period ($\approx$ 25ms). This capability to compute collision distances for a large number of joint postures all at once represents a significant advantage, especially when implemented in sampling-based path planning methods like RRT, and it can also be effectively applied to advanced control strategies such as Model Predictive Control (MPC).
>
> In summary, similar to traditional non-learning methods, PairwiseNet calculates collision distance in a pairwise manner, which is key to its demonstration of high accuracy for systems with high degrees of freedom. Unlike these traditional methods, PairwiseNet offers the advantages of fast inference speed and differentiability, akin to existing data-driven methodologies. Once again, we thank you for your insightful comments, and we are pleased to have the opportunity to clarify our work through your remarks.

---

> > ### Comment · Reviewer_fxzY · 2023-08-15
> > **Thanks for your response**
> >
> > The advantages of learning-based collision distance calculation are already known, and the advantages of computational complexity through simple batch calculation are also mentioned in other papers such as GraphDistNet. As a result, the mention of efficiency through batch computation is not an effective novelty.
> >
> > Therefore, if you have the same or similar computational efficiency without batching and higher efficiency with batching, you may have a stronger claim.

---

> > > ### Author Response · Authors · 2023-08-15
> > > **Response to Reviewer fxzY**
> > >
> > > Thank you for your insightful comment. As you have correctly pointed out, the advantages of reducing computational complexity through simple batch calculation have already been mentioned by other existing learning-based collision distance estimation methodologies. Therefore, it is only natural that this is not the main contribution of our proposed PairwiseNet.
> > >
> > > In Appendix B.1, we have included measurements of inference time, comparing PairwiseNet with non-learning collision distance methodologies, as part of our sincere effort to address your comments. Through this comparison, we would like to emphasize that PairwiseNet can perform efficient inference equivalent to existing learning-based methodologies while calculating collision distance pairwise. This efficiency is notably greater than non-learning collision distance calculation methods like GJK. However, we would like to clarify that this efficient inference is not our major contribution, but rather an aspect we thought relevant to your initial comments.
> > >
> > > The main novelty of PairwiseNet lies in its greatly surpassing existing learning-based methods in collision distance estimation performance in high-dof systems. This has been demonstrated through various performance metrics in Table 1 and can also be confirmed through Figure 5 and the supplementary video. Additionally, PairwiseNet has the added novelty of being able to use the model as-is without additional training, even if a robotic arm is added or its position is changed, thanks to its feature of estimating the collision distance pairwise.
> > >
> > > If our previous communication seemed unclear, please accept our sincere apologies. We hope this response better articulates our position, and we remain open to further discussion and clarification.

---

### Official Review · Reviewer_uPAS · 2023-07-20

**Confidence:** 4
**Originality:** Good
**Technical Quality:** Very Good
**Clarity Of Presentation:** Very Good
**Impact:** 4

**Recommendation:**

Weak Accept: I recommend accepting the paper, but will not argue for my recommendation if the majority of other reviewers have a different opinion.

**Review:**

The major strengths of these work are:
- Computation of collision distance as a pairwise problem for multi-body system is intuitive. This is how even deterministic method of computation of collision distance is performed.
- Separation of encoder and the regressor in the system architecture is useful in reducing training effort especially in multi-robot scenarios.
- The authors provide comparative evaluation with other learning based methods such as ClearanceNet.
- The authors also provide a variety of evaluation metrics and demonstrate the performance for high-dimensional robotic system (28-dof).
- The authors provide a real-world demonstration of their framework.

The areas of improvements are:
- Clearly state the changes in the system the model can handle without the need for retraining.
- Empirical comparison with GraphDistNet
- Variety in the obstacles and multi-robot systems by changing distance of the bases and workspace obstacles of different shapes other than rectangular and their impact on the training effort.
- Inclusion of other learning based collision detection work in related work. See the list in questions for rebuttal.

**Quality Of The Limitations Section:**

Limitations are addressed clearly

**Questions For Rebuttal:**

- The authors should clearly state or experimentally demonstrate how similar the shape elements need to be in the training set to avoid retraining. Can a model trained with UR3 be used to problems using UR5? The shapes are similar but differ in dimensions. Does the model needs complete retraining or pre-trained models can be used with additional layers? This would also clearly indicate what ``slight changes'' the model can handle without retraining.
- As retraining is required if the system contains dissimilar objects from training set, the authors should evaluate the training set volume required for planning problems including variety of shape elements such as multi-robot system with different robots and workspace obstacles not rectangular in shape.
- Although the authors have compared their work with other notable work in clearance estimation, the authors should provide empirical comparison with GraphDistNet, ref. 14. GraphDistNet also decomposes multi-body object in form of a graph to apply it to learning based model.
- The authors show the experimental analysis with multi-robot systems but the multi-robot systems do not include any workspace obstacles and the robot bases are equidistant to each other. The authors should include varying workspace obstacles in the evaluation and vary the distances of the bases. The authors need to also state whether any prior decomposition of the shape obstacles in the training set is required as a pre-processing step.
- The authors should cite and consider the following works:
 Y. Han, W. Zhao, J. Pan and Y. -J. Liu, "Configuration Space Decomposition for Learning-based Collision Checking in High-DOF Robots," 2020 IEEE/RSJ International Conference on Intelligent Robots and Systems (IROS), Las Vegas, NV, USA, 2020, pp. 5678-5684, doi: 10.1109/IROS45743.2020.9341526.
Tan, Qingyang, Zherong Pan, and Dinesh Manocha. "Lcollision: Fast generation of collision-free human poses using learned non-penetration constraints." Proceedings of the AAAI Conference on Artificial Intelligence. Vol. 35. No. 5. 2021.
J. Muñoz, P. Lehner, L. E. Moreno, A. Albu-Schäffer and M. A. Roa, "CollisionGP: Gaussian Process-Based Collision Checking for Robot Motion Planning," in IEEE Robotics and Automation Letters, vol. 8, no. 7, pp. 4036-4043, July 2023, doi: 10.1109/LRA.2023.3280820.

**Robotics Focus:**

Sufficient demonstration on hardware

**Summary Of Paper:**

The paper presents a learning based method for collision distance prediction and its' application in collision detection for motion planning of manipulators. The algorithm uses a pairwise collision distance computation for multi-part objects to address the computational complexity of high-dimensional systems. The learning architecture uses an encoder to compute the shape feature vector from the point cloud of each shapes and then a regressor that uses the shape feature vector along with the transformation between the pairs of geometric shape to predict the minimum distance between them. Experiments evaluate the performance of 7-dof panda robot with varying the number of robots and distance between them in each environment. The performance show the proposed method excels in accuracy, safe-FPR, MSE and AUROC when compared to bounding volume method, Joint and position nn, joint nerf, clearance net and Diffco.


**Summary Of Recommendation:**

Overall, the paper presents an interesting idea of computing collision distance between objects by decomposing it into pairwise distance computation. This is evaluated for high-dimensional systems and compared with other known collision distance method computations.

---

> ### Author Response · Authors · 2023-08-12
> **Response to Reviewer uPAS (3/3)**
>
> **Q5.** The authors need to also state whether any prior decomposition of the shape obstacles in the training set is required as a pre-processing step.
>
> **A5.** Thank you for the insightful comment. We apologize for not providing sufficient details regarding the strategy of decomposing the system elements. PairwiseNet is designed to achieve robust collision distance estimation by dividing the robot system into multiple elements and then calculating the pairwise collision distance between these elements. This necessitates the decomposition of the robot system into individual elements as a preliminary step. Although PairwiseNet does not impose any constraints on an element's convexity, its exceptional performance mainly arises from the simplicity of the pairwise collision distance function compared to the global collision function. Therefore, if a single element is complex and highly non-convex, the pairwise collision distance function might become more complicated, potentially degrading PairwiseNet's performance. Thus, there needs to be a balance between the complexity of decomposition and the performance of PairwiseNet. `We have added a more detailed discussion about the decomposition of the shape elements in Appendix A.1.` Again, I appreciate your comment, which led to better clarification of PairwiseNet.
>
> **Q6.** The authors should cite and consider the following works:
> * Y. Han, W. Zhao, J. Pan and Y. -J. Liu, "Configuration Space Decomposition for Learning-based Collision Checking in High-DOF Robots," 2020 IEEE/RSJ International Conference on Intelligent Robots and Systems (IROS), Las Vegas, NV, USA, 2020, pp. 5678-5684, doi: 10.1109/IROS45743.2020.9341526.
> * Tan, Qingyang, Zherong Pan, and Dinesh Manocha. "Lcollision: Fast generation of collision-free human poses using learned non-penetration constraints." Proceedings of the AAAI Conference on Artificial Intelligence. Vol. 35. No. 5. 2021.
> * J. Muñoz, P. Lehner, L. E. Moreno, A. Albu-Schäffer and M. A. Roa, "CollisionGP: Gaussian Process-Based Collision Checking for Robot Motion Planning," in IEEE Robotics and Automation Letters, vol. 8, no. 7, pp. 4036-4043, July 2023, doi: 10.1109/LRA.2023.3280820.
>
> **A6.** Thank you for your suggested references. In response to your feedback, `we have updated Section 2, "Related Works", to include these references.`

---

> > ### Comment · Reviewer_uPAS · 2023-08-14
> > **Thank you**
> >
> > Thank you for the clarifications of queries and the updates. I hope the empirical comparison with graphdistnet is available by camera-ready version. Otherwise I would suggest the authors to state the challenges faced in its' re-implementation or a comparison with a modified implementable version of graphdistnet in appendix.

---

> ### Author Response · Authors · 2023-08-12
> **Response to Reviewer uPAS (2/3)**
>
> **Q3.** Although the authors have compared their work with other notable work in clearance estimation, the authors should provide empirical comparison with GraphDistNet, ref. 14. GraphDistNet also decomposes multi-body object in form of a graph to apply it to learning based model.
>
> **A3.** We appreciate your suggestion to compare our work with GraphDistNet due to its similarity to our approach in clearance estimation. Indeed, GraphDistNet's method of decomposing multi-body objects in the form of a graph for learning-based models makes it a relevant point of comparison. Unfortunately, we encountered a significant obstacle in this endeavor, as there is no open-source code available for GraphDistNet. We attempted to reproduce GraphDistNet using only the explanations provided in the paper, but the advanced graph neural network structures presented challenges that prevented successful reproduction. Recognizing the importance of this comparison, we plan to contact the authors of GraphDistNet to request the source code. Should they provide it, we intend to add the performance comparison before submitting the camera-ready version.
>
> Even without directly comparing the collision distance estimation performance, we can highlight the novelty of PairwiseNet relative to GraphDistNet. A key difference lies in scalability to high-dof systems. GraphDistNet's performance was only verified in a system with a maximum of 7-dof, and that was within a synthetic planar robot system where a simple graph representation was possible. Additionally, experiments with real robots were only conducted with 3-dof. Since GraphDistNet employs a complex GNN structure that takes the graph itself as the input, the model's inference speed is inevitably sensitive to the complexity of the graph. In examining the experimental results, we observed that the inference speed slows down as the graph becomes more complex, being up to 120 times slower than ClearanceNet. In contrast, PairwiseNet maintains inference speeds comparable to ClearanceNet even in complex systems with 28 DOF across 4 robot arms.
>
> However, as you have suggested, we acknowledge the importance of an empirical comparison with GraphDistNet. We believe that this comparison, if feasible, would greatly enhance our work. Once again, we thank you for highlighting this significant aspect, and we assure you that we will make every effort to address it.
>
> **Q4.** The authors show the experimental analysis with multi-robot systems but the multi-robot systems do not include any workspace obstacles and the robot bases are equidistant to each other. The authors should include varying workspace obstacles in the evaluation and vary the distances of the bases.
>
> **A4.** Thank you for your constructive feedback, which has directed our attention to an essential aspect of our experimental analysis. In response to your suggestion, `we have added the evaluation results of PairwiseNet in multi-arm robot systems with various base positions in Appendix B.4.` In this new evaluation, we have ensured a greater variety in the relative positions between the robot arms. Unlike the previously used three robot arm systems where bases were equidistant, the robot arms in our new systems are arranged in a more irregular and complex manner.
>
> We observed that PairwiseNet successfully estimated the collision distance of these complex multi-arm robot systems. This enhanced analysis not only addresses your concern but also contributes to a deeper understanding of how PairwiseNet performs in more intricate scenarios with varying distances between bases. Thank you once again for this valuable insight, which has significantly contributed to strengthening our research.

---

> ### Author Response · Authors · 2023-08-12
> **Response to Reviewer uPAS (1/3)**
>
> **Q1.** The authors should clearly state or experimentally demonstrate how similar the shape elements need to be in the training set to avoid retraining. Can a model trained with UR3 be used to problems using UR5? The shapes are similar but differ in dimensions. Does the model needs complete retraining or pre-trained models can be used with additional layers? This would also clearly indicate what "slight changes" the model can handle without retraining.
>
> **A1.** Thank you for pointing out this critical aspect, and we apologize that our original manuscript did not clearly state the generalizability of PairwiseNet. The generalizability we proposed is specifically aimed at scenarios where the shape element within the system remains unchanged. In other words, the generalization applies only to systems in which the shape element learned from the training data persists, and where changes might occur in the robot's base position or the obstacle's position, or if known shape elements are newly added. The examples you mentioned, such as UR3 and UR5, encompass generalization across various element shapes, which represents a more complex issue and, unfortunately, is not addressed by PairwiseNet. We cannot guarantee the performance of PairwiseNet if new elements with unknown (untrained) shapes are provided. `We have updated the limitations section in Section 5 to more clearly state these discussions.`
>
> However, generalizing to scenarios where the position of objects within the system has changed (although this is simpler than generalizing across different element shapes) remains a significant challenge for many data-driven methodologies. Even minor adjustments to the position of obstacles or the robot's base can lead to dramatic alterations in the shape and topology of the collision-free space within the joint configuration space. Such changes often force many methods to restart the entire process, from data collection to model training. The novelty of PairwiseNet lies in its ability to be applied immediately in such cases, without the need for retraining.
>
> Certainly, the generalization across various element shapes that you mentioned is a valuable direction for future work, and it could be attempted using a dataset composed of sufficiently diverse shapes. If successful, this would enable more efficient applications of PairwiseNet in a broader range of environments, greatly enhancing the impact of the algorithm. Thank you once again for your valuable comment, and your insight has provided us with an excellent direction for future work.
>
> **Q2.** As retraining is required if the system contains dissimilar objects from training set, the authors should evaluate the training set volume required for planning problems including variety of shape elements such as multi-robot system with different robots and workspace obstacles not rectangular in shape.
>
> **A2.** Thank you for your insightful comment. You are correct in pointing out the need to retrain the model if the system contains dissimilar objects from the training data points. This is indeed a critical aspect of PairwiseNet's application, and we acknowledge this necessity. So, to further address your concern, `we have added an analysis of the training complexity of PairwiseNet in Appendix B.3`, where we explore the specifics of training with different system configurations.
>
> For the multi-arm robot system, we additionally experimented with training PairwiseNet with fewer data points (1 million) and fewer gradient steps. We found that the training time decreased accordingly, but the performance was still compatible with the original PairwiseNet that was trained with more data. This indicates a level of efficiency in training the model even with reduced data.
>
> Moreover, we constructed a two-arm system integrated with some non-rectangular objects (household objects) and trained PairwiseNet in this more complex environment. In this experiment, we used 1 million data points and fewer gradient steps than the original PairwiseNet. Despite these constraints, the model was well-trained, confirming the robustness of PairwiseNet in handling diverse shapes within the workspace. This expanded analysis aims to provide a comprehensive understanding of the model's applicability in various scenarios, as you rightly suggested. Thank you once again for bringing this to our attention.

---

### Official Review · Reviewer_xJsm · 2023-07-22

**Confidence:** 4
**Originality:** Fair
**Technical Quality:** Good
**Clarity Of Presentation:** Very Good
**Impact:** 3

**Recommendation:**

Weak Accept: I recommend accepting the paper, but will not argue for my recommendation if the majority of other reviewers have a different opinion.

**Review:**

This paper presents a novel approach and demonstrates improvement over a comprehensive set of baselines.

Given that the intended use case for this is motion planning, it would be good to see some speed comparisons between a standard collision checking approach and this network.

Additionally, this approach doesn't extend to the types of problems that collision detection is most often needed for (detecting collisions between the robot and an arbitrary environment geometries).

**Quality Of The Limitations Section:**

Limitations are addressed clearly

**Questions For Rebuttal:**

This approach consumes more GPU memory than the global approaches because the link distances have to be computed in parallel. If GPU memory isn't a bottleneck, couldn't you just compute pairwise distances between every pair of points in the pointclouds of the two robots in parallel? This would result in perfect performance and generalization.

**Robotics Focus:**

Highly relevant to robotics but no hardware experiments

**Summary Of Paper:**

This paper proposes an architecture for estimating the distance between links in a robot model. The key novelty of this paper is that they split up the global robot model into links and link poses and estimate the pairwise distance between links.

**Summary Of Recommendation:**

This paper presents a novel approach for collision distance estimation and demonstrates improvement over a comprehensive set of baselines. The paper would benefit from some experiments regarding space and memory tradeoffs between learned and classic collision detection in motion planners.

---

> ### Author Response · Authors · 2023-08-12
> **Response to Reviewer xjsm (2/2)**
>
> **Q3.** This approach consumes more GPU memory than the global approaches because the link distances have to be computed in parallel. If GPU memory isn't a bottleneck, couldn't you just compute pairwise distances between every pair of points in the pointclouds of the two robots in parallel? This would result in perfect performance and generalization.
>
> **A3.** Thank you for your insightful comment regarding GPU memory consumption and the possibility of computing pairwise distances between every point in the pointclouds in parallel. I would like to address your comment by elaborating on three main aspects of PairwiseNet:
>
> * **GPU Memory Consumption** : Your observation is correct that computing the link distances in parallel might generally lead to higher memory consumption compared to global approaches. However, we have designed PairwiseNet to not require significant GPU memory (only 67,585 parameters for the regressor network and the shape feature vectors, totaling 270kB). This is achieved by encoding each point cloud data into a 32-dimensional feature vector and saving it, eliminating the need to save all the point cloud data. Conversely, computing pairwise distances between every pair of points in the point clouds would necessitate storing all the point cloud data in GPU memory, leading to higher memory consumption. Therefore, PairwiseNet is more memory-efficient in comparison.
> * **Inference Complexity** : It is important to note that the inference complexity of PairwiseNet is independent of the number of points in the point cloud data. While the idea of computing pairwise distances between every pair of points in the point clouds might seem appealing, it would indeed increase the computational complexity toThis quadratic growth in complexity would make the method less scalable and efficient, especially for large point clouds. In our experiments, in a situation where each point cloud contains only 100 points, computing pairwise distances between all points in the point cloud takes 2-3 times longer than PairwiseNet, even with the use of GPU. This clearly shows how PairwiseNet offers a significant advantage in computational efficiency, particularly as the point cloud size grows. $O(M^2)$ as the number of points in the point cloud $M$ increases.
> * **Inference Time with CPU Calculation** : Another advantage of PairwiseNet is its short inference time, even when computed using a CPU. Calculating pairwise distances between all points in the point cloud would make this efficiency almost impossible, particularly without the support of a powerful GPU.
>
> In summary, PairwiseNet offers an efficient and scalable solution that does not suffer from high memory consumption and complexity. The approach you suggested would indeed provide perfect performance and generalization in a scenario where GPU memory and computational power are not constraints. However, PairwiseNet aims to provide a more practical and versatile solution that balances accuracy, computational efficiency, and memory requirements. Thank you once again for your valuable suggestion, and we are pleased that your comment has created an opportunity to convey the novelty of PairwiseNet more clearly.

---

> ### Author Response · Authors · 2023-08-12
> **Response to Reviewer xjsm (1/2)**
>
> **Q1.** Given that the intended use case for this is motion planning, it would be good to see some speed comparisons between a standard collision checking approach and this network.
>
> **A1.** Thank you for this suggestion. `We have added the results comparing the inference time complexity of PairwiseNet with the standard collision distance calculation method, the GJK algorithm, in Appendix B.1.` The inference time for each method is measured over 1000 repeated collision distance calculations, and PairwiseNet significantly outperforms the computation speed of the GJK algorithm when using both CPU and GPU, thanks to the efficient batch computation of the neural network. Moreover, PairwiseNet can input many joint postures in a batch to calculate collision distances all at once, and in this case, it is possible to calculate the collision distances for 1000 joint postures in a very short time. This ability to calculate collision distances of numerous joint postures all at once in a short time becomes a significant advantage when utilized in sampling-based path planning methods like RRT, and it can also be efficiently applied to control techniques such as Model Predictive Control (MPC). Additionally, in the case of reactive path planning or optimization-based path planning methodologies, the derivatives of collision distances with respect to joint configurations are mostly needed, and PairwiseNet can also calculate this in a short time.
>
> **Q2.** this approach doesn't extend to the types of problems that collision detection is most often needed for (detecting collisions between the robot and an arbitrary environment geometries).
>
> **A2.** Thank you for pointing out this important issue. As you mentioned, estimating the collision distance in an arbitrary environment is a meaningful problem, and one that we have not explicitly addressed in this paper. There are a number of ways to generalize our method to this problem. For example, if we include data for estimating collision distances between appropriate shape primitives (such as spheres, cylinders, superquadrics, etc.), one possible approach to use PairwiseNet to estimate collision distances between arbitrary shapes is to approximate these shapes by a collection of these primitive shapes. We believe this is a suitable topic for further investigation, but the focus of this paper remains on self-collisions for complex high-dof systems, which itself is an important and increasingly relevant problem for collaborative robots engaged in interaction tasks.

---

### Decision · Program_Chairs · 2023-08-30

**Decision:**

Accept (Poster)

**Comment:**

The reviewers find this work to be both interesting and novel. The comparison to GJK nicely shows the improvement of using the proposed approach over common analytic approaches to collision checking.

I generally agree with the reviewers that comparing to GraphDistNet would improve the paper and I thank the authors for their attempts to do so and understand the difficulties in making this happen. If this could be included for the final version of the paper it would definitely increase its value and quality.